# Trends in Incidence and Mortality of Primary Liver Cancer in Lithuania 1998–2015

**DOI:** 10.3390/ijerph18031191

**Published:** 2021-01-29

**Authors:** Audrius Dulskas, Povilas Kavaliauskas, Kestutis Zagminas, Ligita Jancoriene, Giedre Smailyte

**Affiliations:** 1Laboratory of Clinical Oncology, National Cancer Institute, 1 Santariskiu Str., LT-08406 Vilnius, Lithuania; 2Clinic of Internal Medicine, Faculty of Medicine, Family Medicine and Oncology Institute of Clinical Medicine, Vilnius University, M. K. Čiurlionio Str. 21/27, LT-03101 Vilnius, Lithuania; 3Department of Public Health, Faculty of Medicine, Institute of Health Sciences, Vilnius University, LT-03101 Vilnius, Lithuania; Povilaskava@gmail.com (P.K.); kestutis.zagminas@mf.vu.lt (K.Z.); giedre.smailyte@nvi.lt (G.S.); 4Center of Infectious Diseases, Vilnius University Hospital Santaros Klinikos, LT-08406 Vilnius, Lithuania; Ligita.Jancoriene@santa.lt; 5Clinic of Infectious Diseases and Dermatovenerology, Faculty of Medicine, Institute of Clinical Medicine, Vilnius University, LT-03101 Vilnius, Lithuania; 6Laboratory of Cancer Epidemiology, National Cancer Institute, LT-08406 Vilnius, Lithuania

**Keywords:** primary liver cancer, hepatocellular carcinoma, intrahepatic cholangiocarcinoma, incidence, mortality, age-standardized rates

## Abstract

*Background*: Recently, reports have suggested that rates of liver cancer have increased during the last decades in developed countries; increasing hepatocellular carcinoma and cholangiocarcinoma rates were reported. The aim of this study was to examine time trends in incidence and mortality rates of liver cancer for the period of 1998–2015 in Lithuania by sex, age, and histology. *Methods*: We examined the incidence of liver cancer from 1998 to 2015 using data from the Lithuanian Cancer Registry. Age-standardized incidence rates were calculated by sex, age, and histology. Trends were analyzed using the Joinpoint Regression Program to estimate the annual percent change. *Results*: A total of 3086 primary liver cancer cases were diagnosed, and 2923 patients died from liver cancer. The total number of liver cancer cases changed from 132 in 1998 to 239 in 2015. Liver cancer incidence rates changed during the study period from 5.02/100,000 in 1998 to 10.54/100,000 in 2015 in men and from 2.43/100,000 in 1998 to 6.25/100,000 in 2015 in women. Annual percentage changes (APCs) in the age-standardized rates over this period were 4.5% for incidence and 3.6% for mortality. Hepatocellular cancer incidence rates were stable from 1998 to 2005 (APC −5.9, *p* = 0.1) and later increased by 6.7% per year (*p* < 0.001). Intrahepatic ductal carcinoma incidence increased by 8.9% per year throughout the study period. The rise in incidence was observed in all age groups; however, in age groups < 50 and between 70 and 79 years, observed changes were not statistically significant. For mortality, the significant point of trend change was detected in 2001, where after stable mortality, rates started to increase by 2.4% per year. *Conclusions*: Primary liver cancer incidence and mortality increased in both sexes in Lithuania. The rise om incidence was observed in both sexes and main histology groups. The increasing incidence trend may be related to the prevalence of main risk factors (alcohol consumption, hepatitis B and C infections. and diabetes).

## 1. Introduction

Primary cancer of the liver is the sixth most common cancer in the world and the second most common cause of cancer mortality [1]. Incidence and mortality patterns, due to the poor prognosis of liver cancer, are close and almost equivalent in world regions. In 2018, the estimated global incidence rate of liver cancer per 100,000 person-years was 9.3, while the corresponding mortality rate was 8.5 [1]. Primary liver cancer includes hepatocellular carcinoma (comprising 75–85% of cases) and intrahepatic cholangiocarcinoma (10–15% of cases), as well as other rare types of liver cancer [2,3].

Previous studies have demonstrated that the incidence of liver cancer varies significantly around the world. According 2018 GLOBOCAN estimates, the highest incidence rates in men and women were reported in Asia and Africa, with the highest incidence estimated in Mongolia (93.7/100,000) [1]. The overall burden from liver cancer is most pronounced in transitioning countries, while the lowest incidence and mortality rates occur in regions with a high income [4,5].

In recent years, substantial changes have been reported in the incidence of liver cancer. A decreasing trend has been reported in Asian counties with high liver cancer incidence, while rising incidence rates have been reported in the developed countries, including Western Europe, USA, and Australia [4,5,6,7,8,9,10,11,12]. Liver cancer variability is likely due to the prevalence of risk factors between the regions and specific population groups. Known risk factors of liver cancer include chronic infection with hepatitis B virus (HBV) and hepatitis C virus (HCV), aflatoxins, alcohol and smoking, obesity, diabetes, and metabolic syndrome [12,13,14].

The most pronounced trends in increased liver cancer incidence have been observed in some European countries, where incidence rates are still relatively low. Lithuania is a country with a low incidence of liver cancer; however, to the best of our knowledge, incidence rates of primary liver cancer and histology-specific liver cancer rates have not been reported to date.

The purpose of this study was to examine the time trends in the incidence and mortality rates of liver cancer for the period of 1998–2015 in Lithuania by sex, age, and histology.

## 2. Material and Methods

Ethical approval for the analysis of the population-based cancer registry data was not required.

The study was based on the Lithuanian Cancer Registry database, covering a population of around 3 million residents, according to the 2011 census. The main sources of data were notifications gathered from all hospitals and diagnostic centers in Lithuania. The data from the Lithuanian Cancer Registry database is publicly available. Additionally, death certificate information and population registry information, which verify vital status, are available. The study was based on all cases of primary liver cancer (International Classification of Disease, Tenth Revision (ICD-10) C220 and C221) reported to the Registry during 1998–2015. For the analyses, patients were categorized by age at diagnosis (0–14, 15–49, and 50+ years), tumor histology, and stage of the disease. The histology of the tumors was coded according to International Classification of Diseases for Oncology (ICD-O-3). The histology codes were grouped into the following categories: hepatocellular carcinoma (ICDO-3 codes: 8170–8175), cholangiocarcinoma (ICDO-3 codes: 8032–8033, 8041, 8050, 8070–8071, 8140–8141, 8160, 8260, 8480, 8481, 8490, and 8560), other, and unspecified.

Age-specific and age-standardized incidence rates were calculated. Standardization was performed using the direct method (European standard population). Age-standardized rates were calculated for all ages combined, for histology categories, and for age groups. Corresponding population data, by age, sex, and year, were available from Statistics Lithuania.

A joinpoint regression model was used to provide the annual percentage changes (APCs) and to detect points in time where statistically significant changes in the trends occurred. The joinpoint regression analysis identifies the best-fitting points (joinpoints) where a significant change in the linear slope (on a log scale) of the trend is detected. The tests of significance use a Monte Carlo permutation method [15]. Annual percent changes were considered statistically significant if *p* was <0.05. Joinpoint analysis was performed for all ages combined and for the age groups, <50, 50–59, 60–69, 70–79, and 80 years or more. A maximum number of three joinpoints was allowed for estimations. We defined the minimum number of observations from a joinpoint to either end of the data as three, and the minimum number of observations between the two joinpoints as two (excluding any joinpoint that falls on an observation). Joinpoint software (version 4.8.0.1) (IMS, Inc. under contract for the National Cancer Institute, Bethesda, MD, USA) was used [16].

## 3. Results

During the study period in Lithuania, 3086 primary liver cancer cases were diagnosed and 2923 patients died from liver cancer. Detailed characteristics of the study groups are shown in Table 1. The total number of liver cancer cases changed from 132 in 1998 to 239 in 2015. Hepatocellular carcinoma was the predominant histology (1394 cases), comprising 45.2% of all liver cancers, followed by intrahepatic cholangiocarcinoma (17.8%, 548 cases). The remaining 7% (215 cases) with a specified histology were neuroendocrine tumors, hepatoblastomas, or sarcomas. The proportion of primary liver tumors with unreported histology was 30.1% (929 cases). From 1998 to 2007, incidence and mortality rates of primary liver cancer were close, at approximately 4/100,000 (Figure 1). From 2002 onward, the incidence rate became slightly higher than the mortality rate. During the study period, the incidence rate increased to 5.70/100,000 and mortality to 4.76/100,000 in 2015. During the period of 1998–2015, liver cancer incidence rates were considerably lower for women than for men, with rates in men almost twice as high as in women. Liver cancer incidence rates changed during the study period from 5.02/100,000 in 1998 to 8.94/100,000 in 2015 in men, and from 2.43/100,000 to 3.48/100,000 in women. (Figure 2).

Incidence rates of primary liver cancer by histology are shown in Figure 3. In both sexes, hepatocellular carcinoma and cholangiocarcinoma incidence increased, which, in turn, increased the incidence of primary liver cancer with unspecified histology. In the group of others, specified histology was also observed.

The results of the joinpoint analysis of incidence and mortality time trends are shown in Table 2. Annual percentage changes in the age-standardized rates over this period were 4.5% for incidence and 0.5% for mortality. In men and women, incidence was increasing steadily during the entire study period (3.1% and 2.1% per year, respectively). For both main histology groups, increasing incidence was observed. Hepatocellular cancer incidence rates were stable from 1998 to 2005 (APC −5.9, *p* = 0.1) and later increased by 6.7% per year (*p* < 0.001). Intrahepatic ductal carcinoma incidence was increasing by 8.9% per year during the study period. The rise of incidence was observed in all age groups; however, in age groups <50 and 70–79 years, the observed changes were not statistically significant (Figure 4). For mortality, the significant point of trend change was detected in 2001; after stable mortality, rates started to increase by 2.4% per year.

## 4. Discussion

This paper presented, for the first time, both the mortality and incidence rates of primary liver cancer and showed that they rose in Lithuania from 1998 to 2015. The rise in incidence was observed in both sexes, both main histology groups, and in all age groups. Primary liver cancer was more common among men than women and the rate of increase in the incidence was greater for cholangiocarcinoma than for hepatocellular carcinoma.

The increasing incidence of liver cancer in Lithuania confirms and expands the previous findings on hepatocellular carcinoma and cholangiocarcinoma trends reported in low endemic areas [4,5,6,7,8,9,10,11]. Hepatocellular carcinoma is the most common form of liver cancer; its incidence remains alarmingly high in the developing world and is steadily rising across most of the developed world [4,6,9]. Changes in the prevalence of the two major risk factors, HBV and HCV infection, is hypothesized to be responsible for decreasing incidence in developing countries, while obesity and diabetes may be factors influencing the increases in the developed world [6,9].

According to the results from the Global Burden of Disease Study, globally, in 2016, more than 40% of liver cancer was caused by HBV, followed by other causes (23.3%), HCV (18.7%), and alcohol consumption (14.7%) [5]. In Eastern Europe, the proportion of liver cancer due to HBV slightly decreased from 39.6% in 1990 to 36.3% in 2016, while the proportion of liver cancer due to HCV was relatively stable (11.9% in 1990 and 12.1% in 2016).

Hepatocellular carcinoma (HCC) is a predominant form of primary liver cancer. In Lithuania during the study period, the proportion of HCC accounted for 45.2% of all primary liver cancer cases and the incidence significantly increased from 1998 to 2015. HCC incidence is related to the major HCC risk factors. HBV and HCV still are the most important risk factors for HCC.

### 4.1. HBV Infection

Over 400 million people chronically infected with HBV are at high risk of developing liver cirrhosis and hepatocellular carcinoma, making HBV the most common carcinogen after tobacco [17]. The HBV preferentially infects hepatocytes and can establish a chronic infection within the liver. Once chronic infection is established, complete eradication of the virus is still not possible, and these patients face the risk of HCC development. The lifetime risk of developing HCC is 10- to 25-fold greater for chronic HBV carriers in comparison to non-infected populations [17].

A strong geographic correlation has been demonstrated between the prevalence of chronic HBV infection and the incidence of HCC [18,19]. The prevalence of chronic HBV infection varies strikingly in different geographic areas and in different populations, with national prevalence ranging from 0.1% to 35% [20]. In Lithuania, prevalence of hepatitis B surface antigen (HBsAg) in different age groups ranged from 1.1% in the 20–29 years group (95% CI 0.02–5.9) to 1.5% in the 40–49 years group (95% Confidence interval (CI 0.3–3.8), among the adult general population in 1996 [21]. The prevalence of HBsAg per 100 first-time non-remunerated donations at the National Blood Center in Lithuania in 2005 was estimated at 1.58% (95% CI 1.25–1.97) [22]. In Lithuania, acute hepatitis B has been a reportable disease since 1965. The incidence of acute HBV infection varied significantly during the period of 1965 to 2015. From 1965 to 1980, the incidence rate (cases per 100,000 population) increased from 10.6 in 1965 to 54.1 in 1980. It is likely that the increase in the reportable incidence may have been mainly due to improvements in surveillance and specific laboratory diagnostics. Since 1980, the incidence has been steadily declining. In Lithuania, the highest incidence of acute hepatitis B is among young adults, and high-risk sexual activity and injecting drug use account for most cases of newly acquired hepatitis B [23]. Reducing exposures to these risk factors and introducing an HBV immunization program is the most effective way to lower the risk of developing liver cancer. Lithuania started the universal hepatitis B immunization program for neonates in 1998. Vaccination coverage of neonates varied between 93% and 99% [24]. These efforts were significantly associated with the decrease in the incidence of acute hepatitis B. It is still too soon, however, for HBV vaccination to have a significant effect on HCC rates among adults, as the first cohort of vaccinated newborns are only now in their thirties–forties. While the incidence of acute infection is falling, relatively large cohorts of chronically infected adults continue to progress to cirrhosis and liver cancer. Although the HBV vaccine has no effect on established infections, treatment of chronic infections dramatically improved with the introduction of lamivudine (LAM), the first nucleos(t)ide analogue (NA), in 1998. The two most commonly used first-line NAs are now entecavir (ETV) and tenofovir, both introduced in the first decade of the twenty-first century. The risk of progression of chronic HBV infection to cirrhosis and HCC is variable and is affected by the host’s immune response. The incidence of cirrhosis ranges from 8% to 20% in untreated chronic hepatitis B (CHB) patients and, among those with cirrhosis, the 5-year cumulative risk of hepatic decompensation is 20%. The annual risk of HCC in patients with cirrhosis has been reported to be 2–5% [25,26].

The main goal of therapy for patients with chronic HBV infection is to improve survival and quality of life by preventing disease progression, and, consequently, HCC development. In Lithuania for a long time, there were no reimbursements for the first-line NA therapies and a number of LAM-treated patients developed resistance not only to LAM, but later to ETV as well. It could be suggested as one of the main reasons of the increase in HBV-associated HCC cases in Lithuania in the last decade.

During the study period, HCC incidence in Lithuania significantly increased, while the trend in the incidence of acute hepatitis B since 1980 reversed. Changes in the epidemiology of hepatitis B (morbidity trends, structure of risk factors, and relatively low prevalence of HBsAg) and a successful vaccination program suggest that HBV could greatly contribute to the decrease in new acute viral hepatitis B cases as well as HCC cases caused by hepatitis B virus infection in the future. Availability of the first line treatment of patients with chronic hepatitis B could be suggested as one of the main reasons for the increase in HBV-associated HCC.

### 4.2. HCV Infection

Chronic HCV infection accounts for more than a third of all cases of HCC [27]. In contrast to HBV, an infection with HCV becomes chronic in most cases. Chronic HCV infection is often associated with the development of liver cirrhosis, hepatocellular cancer, liver failure, and death. HCV-induced HCC development is a multi-step process that may last 20–40 years and involves chronic hepatic inflammation, progressive liver fibrosis, initiation of neoplastic clones, and tumor progression in a carcinogenic tissue microenvironment. HCV-associated HCC appears to affect older patients and to follow more severe liver disease than HBV-associated HCC [28]. Globally, in 2015, 71 million people were living with chronic HCV infection [29]. There is a distinct geographical variation in HCV prevalence in the European Union and neighboring countries. The prevalence of HCV in the general population ranges from 0.4% in Sweden, Germany, and the Netherlands to over 20% in one region of Italy. In general, countries in the southern parts of European Union have a higher HCV prevalence compared to countries in the north or west of the EU [30].

In Lithuania, anti-HCV prevalence in the general population among adults was found to range from 1.7–2.8% [31,32]. The prevalence of anti-HCV per 100 first-time non-remunerated donations at the National Blood Center in Lithuania in 2005 was estimated at 0.93% (95% CI 0.68–1.24) [22]. In Lithuania, acute hepatitis C has been a reportable disease since 1993. The peak of incidence was observed in 2001 (5.68 cases per 100,000 population). Since 2001, the incidence has been declining [32]. Before initiating screening of blood donors, anti-HCV was found in 7.9% of commercial blood donors, 13.9% of commercial blood plasma donors, 48.3% of hemodialysis patients, and 29.4% of prisoners. In Lithuania, blood transfusion was the most common risk factor for acquiring HCV before the introduction of routine blood screening for anti-HCV and were present in 37.0% of all acute hepatitis cases in 1991–1992 [33]. HCV infection is highly prevalent among injecting drug users (IDUs) in Europe. Anti-HCV prevalence among IDUs has been estimated at 67% worldwide. The recorded midpoint prevalence estimates in Europe range from 21.1% to 90.5%, with approximately half of all countries estimated to have 60% prevalence or higher [34]. In Lithuania, the midpoint estimate of anti-HCV prevalence amongst IDUs was found at 89.4%.

In the past decades, pegylated interferon combined with ribavirin has been used extensively for HCV treatment, and interferon is thought to have an antitumor property. Direct-acting antivirals (DAAs) have fundamentally changed HCV therapy due to their high efficacy and tolerability. Currently available high-efficacy DAAs, mostly short-term (8–12 weeks) and pangenotypic, have been introduced for the treatment of the hepatitis C virus in 2014, and the sustained virological response (SVR) rate after DAAs was reported to be over 96%. Because of the high SVR rate, the risk of HCC was expected to reduce. However, an unexpected high risk of HCC recurrence after DAA treatment was reported, and thus the dispute about the association of DAA and HCC arose. Patients with chronic HCV infection who develop HCC soon after treatment with DAA may have been harboring hidden tumors. They have a lower SVR rate, since active HCC hampers DAA efficacy. Lack of achieving SVR is a strong independent predictor of the development of HCC early after the treatment of hepatitis C with DAA [35,36,37]. Eradication of HCV reduces, but does not eliminate, the risk of HCC development, especially when the advanced hepatic fibrosis already originated.

Treatment standards in Lithuania are following the global trends in terms of used therapies, except that sofosbuvir combinations, the only suitable DAA in case of severe cirrhosis (Child B or C), are not yet reimbursed.

The presented data on HC epidemiology suggest that in Lithuania, there is a large reservoir of chronic HCV infection formed in the period before the blood donor screening for HCV infection, which, among other factors, may lead to an increase in incidence of HCC in the elderly age groups. Currently, the main source of chronic infection reservoir is the acquisition of HCV infection at a young age through the use of injection drugs. HCV transmission can be prevented with injection safety measures, universal precautions, and harm reduction programs for people who inject drugs. Until 2014, HCV infection was difficult to eradicate. With the discovery of sofosbuvir and other anti-viral drugs, almost all HCV infection can be cured [19]. The development of new HCV treatments that can achieve cure rates of over 90% for people with chronic HCV infection has the potential to change the burden of disease caused by HCV infections.

### 4.3. Other Risk Factors

There is sufficient evidence in humans that the carcinogenicity of alcoholic beverages and malignant tumors of the liver are causally related to the consumption of alcoholic beverages [38]. In Western Europe, the proportion of liver cancer due to alcohol consumption decreased from 39.3% in 1990 to 34.5% in 2016, while in Eastern Europe, the proportion of liver cancer due to alcohol consumption increased from 28.7% in 1990 to 35.4% in 2016 [5]. Lithuania is characterized by having one of the most detrimental drinking patterns in the European Union (EU) [39]. Lithuania, with a recorded and unrecorded alcohol consumption of 17.2 L of pure alcohol per year in 2002, ranks second among countries with the highest alcohol intake in Europe [40]. A study of trends in alcohol consumption suggests that alcohol consumption in Lithuania has increased during the post-communist transition period, especially among women. Only a slight decrease in the frequency of regular alcohol drinking was observed between 2008 and 2010 [41]. Despite the reported decreasing alcohol-related mortality in Lithuania, our country remains among the leaders in Europe and the world in alcohol-related mortality [42], and high alcohol consumption in Lithuania could have added to the incidence of hepatocellular carcinoma.

Diabetes mellitus is a risk factor for hepatocellular carcinoma. An analysis of cancer risk among diabetic patients in Lithuania showed a twofold rise in liver cancer incidence among men with diabetes and a 45% rise among women with diabetes mellitus compared to the general population [43]. These findings are similar to those observed in the meta-analysis by Wang et al., where the combined risk estimate for hepatocellular cancer was 2.31 (95% CI 1.87–2.84) among diabetic individuals [44].

Improved medical care of cirrhotic patients may also lead to the increase in HCC [45].

The progressive increase in the incidence and mortality rates of primary liver cancer in Lithuania may reflect either a true increase or an epidemiological artefact. Recent technological advances in the management of liver cancers, including modern hepatobiliary imaging, image-guided biopsies, and widespread availability of magnetic resonance cholangiopancreatography may have contributed to the recent trends. In Lithuania, primary liver cancer is diagnosed using ultrasound, computed tomography (CT) scan, magnetic resonance imaging (MRI), and histological investigation data according to worldwide recommendations. With the increasing usage of CT scan and MRI for liver lesions, the incidence of liver cancer is also increasing. The key treatment options for liver cancer are surgical resection, liver transplant (for patients with cirrhosis and lesion, which subsides to the Milan criteria), radioablation therapy, chemoablation, and chemotherapy. Immunotherapy and gene therapy are being studied and introduced in treating this disease. New biomaterials, such as nanodiamonds and natural polymerparticles, are being introduced and actively investigated in treating primary liver cancer. Another novel minimal invasive technique, radiofrequency ablation of liver lesions, started in Lithuania in 2010 and minimal invasive laparoscopic resections have been performed since 2015.

The major strength of the present study is that we used the whole population Cancer Registry data of Lithuania. We were able, for the first time, to demonstrate trends in both mortality and incidence of primary liver cancer for the entire population of Lithuania. In addition, incidence trends by sex, age, and histology were analyzed. However, due to the limited number of primary liver carcinoma cases in whole population, the trends are estimations subject to error. Histology-specific trends also may be partly influenced by high proportions of liver cancer with unreported histology, and this is the additional limitation of our study.

Nonetheless, in our study, a true increase may underlie the trend in the incidence and mortality being reported. The fact that both incidence and mortality increased supports a non-artefactual effect. Furthermore, our data show that the proportion of cases reported to the cancer registry that had histology unreported has been increasing, suggesting that the observed increase in hepatocellular carcinoma and cholangiocarcinoma incidence rates shows the true increase in incidences.

## 5. Conclusions

Primary liver cancer incidence and mortality increased in both sexes in Lithuania. The rise in incidence was observed in both sexes and both main histology groups. The increasing incidence trend may be related to the prevalence of main risk factors (alcohol consumption, HBV and HCV infections, and diabetes). The observed similar increase in liver cancer incidence and mortality rates suggests that there has not been any improvement in survival from primary liver cancer, despite changes in the diagnosis and treatment of liver cancer during the study period. Follow-up studies of these changing trends in the incidence of liver cancer and mortality need to be closely monitored.

## Figures and Tables

**Figure 1 ijerph-18-01191-f001:**
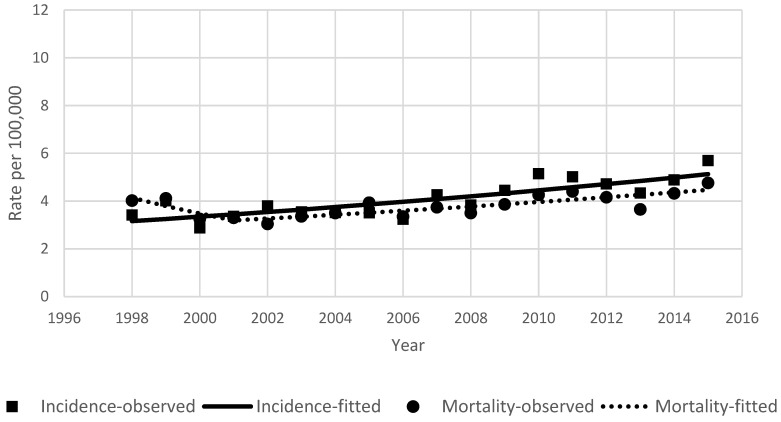
Primary liver cancer in Lithuania between 1998 and 2015. Age-standardized incidence and mortality rates in both sexes.

**Figure 2 ijerph-18-01191-f002:**
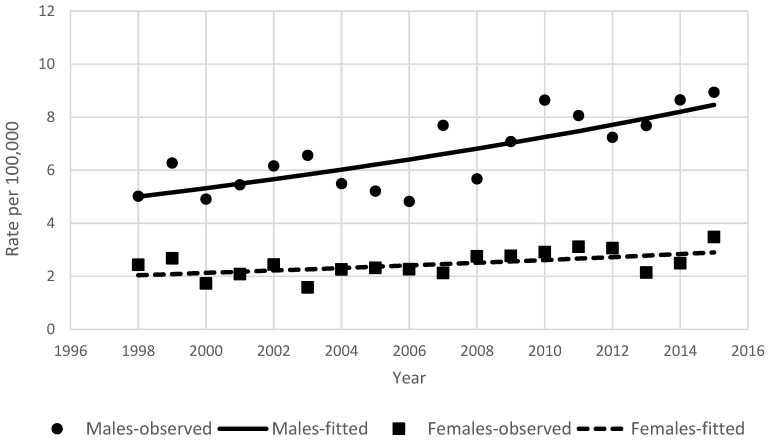
Age-standardized incidence rates of primary liver cancer by sex in Lithuania between 1998 and 2015.

**Figure 3 ijerph-18-01191-f003:**
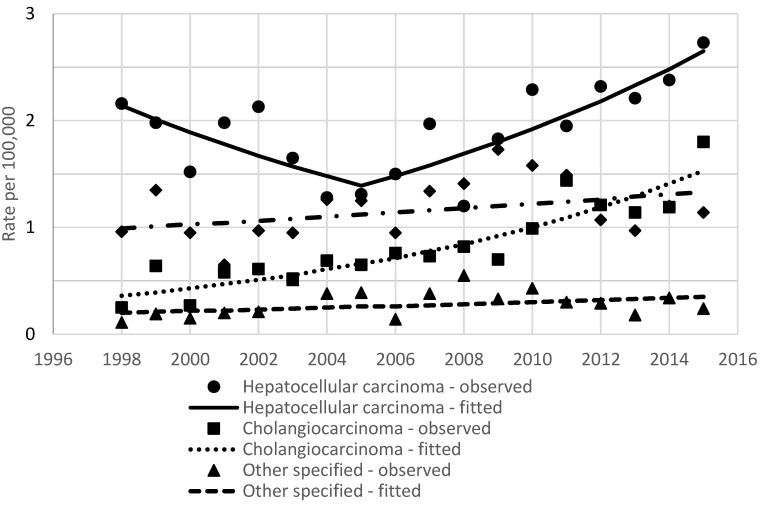
Age-standardized incidence rates of primary liver cancer by histology in Lithuania between 1998 and 2015.

**Figure 4 ijerph-18-01191-f004:**
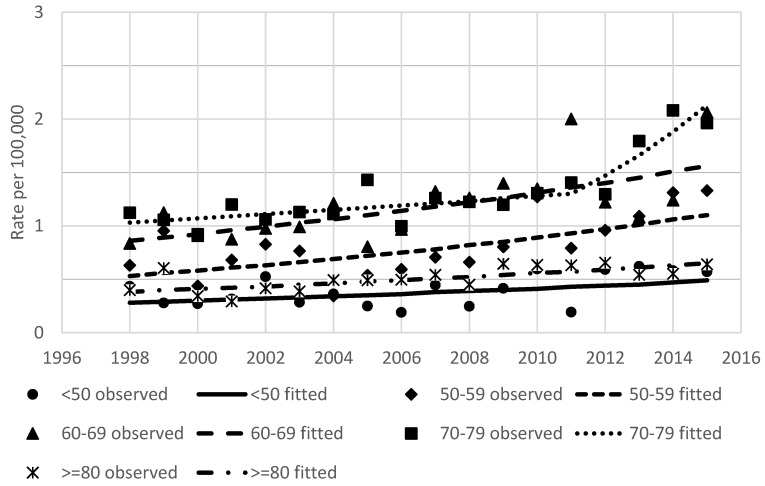
Age-standardized incidence rates of primary liver cancer by age group in Lithuania between 1998 and 2015.

**Table 1 ijerph-18-01191-t001:** Characteristics of the primary liver cancer patients included in the analysis.

Variable	Men		Women		Both Sexes	
	Cases	%	Cases	%	Cases	%
Incidence						
Total	1850	59.95	1236	40.05	3086	100.00
Histology						
Hepatocellular carcinoma	971	52.49	423	34.22	1394	45.17
Cholangiocarcinoma	222	12.00	326	26.38	548	17.76
Other specified	109	5.89	106	8.58	215	6.97
Unspecified	548	29.62	381	30.83	929	30.10
Age group (years)						
<50	148	8.00	82	6.63	230	7.45
50–59	335	18.11	127	10.28	462	14.97
60–69	560	30.27	236	19.09	796	25.79
70–79	591	31.95	474	38.35	1065	34.51
≥80	216	11.68	317	25.65	533	17.27
Mortality	1702	58.23	1221	41.77	2923	100.00

**Table 2 ijerph-18-01191-t002:** Changes in age-standardized primary liver cancer incidence and mortality in Lithuania between 1998 to 2015.

	1998		2015		APC *	*p*	Trend 1	APC	*p*	Trend 2	APC	*p*
Variable	Cases	Rate	Cases	Rate	1998–2015							
Incidence												
Total	132	3.42	239	5.7	4.5	<0.001	–	–	–	–	–	–
SEX												
Male	72	5.02	141	8.94	3.1	<0.001	–	–	–	–	–	–
Female	60	2.43	98	3.48	2.1	<0.001	–	–	–	–	–	–
Histology												
Hepatocellular carcinoma	83	2.16	110	2.73	1.3	<0.001	1998–2005	−5.9	0.1	2005–2015	6.7	<0.001
Cholangiocarcinoma	9	0.25	76	1.8	8.9	<0.001	–	–	–	–	–	–
Other specified	3	0.11	9	0.24	3.3	<0.001	–	–	–	–	–	–
Unspecified	36	0.96	44	1.14	1.8	<0.001	–	–	–	–	–	–
Age group (years)												
<50	14	0.43	16	0.57	3.3	0.11	–	–	–	–	–	–
50–59	20	0.63	45	1.33	4.3	<0.001	–	–	–	–	–	–
60–69	34	0.83	73	2.06	3.6	<0.001	–	–	–	–	–	–
70–79	47	1.12	57	1.09	0.8	<0.2	–	–	–	–	–	–
≥80	17	0.40	48	0.64	3.1	<0.001	–	–	–	–	–	–
Mortality	156	4.02	210	4.76	0.5	<0.001	1998–2001	−8.4	0.1	2001–2015	2.4	<0.001

* APC—annual percentage changes.

## Data Availability

Data are available upon reasonable request.

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
