# Peer review of "Trends in Incidence and Mortality of Primary Liver Cancer in Lithuania 1998–2015"

_ijerph, 2021, doi:10.3390/ijerph18031191_

Round 1

Reviewer 1 Report

Using data from the National cancer registry, the authors provide findings on the time trend of primary liver cancer incidence and mortality in Lithuania over the period 1998-2015.

The paper is well performed generating useful information even for foreigner Readers. I have few comments:

  • The raising incidence rate of hepatocellular carcinoma (HCC) in developped world is mostly due to improvements in the medical management of non-HCC related complications of liver cirrhosis. It has led to a longer survival of patients with cirrhosis, who are at greater risk over time of HCC development. As consequence ,the tumour has become the first complication and the leading cause of mortality among cirrhotics. The authors should acknowledge these concepts.
  • Discussion is too long. Several paragraphs can be deleted.

Author Response

First round

A point-by-point response to the reviewer's comments:

Dear Reviewers,

Thank you for your letter and constructive comments concerning our manuscript entitled “Trends in incidence and mortality of primary liver cancer in Lithuania 1998-2015”. The paper was revised substantially. Following changes have been made. They are as follows, revised paragraphs, sentences, words are below:

Reviewer #1: The raising incidence rate of hepatocellular carcinoma (HCC) in developped world is mostly due to improvements in the medical management of non-HCC related complications of liver cirrhosis. It has led to a longer survival of patients with cirrhosis, who are at greater risk over time of HCC development. As consequence ,the tumour has become the first complication and the leading cause of mortality among cirrhotics. The authors should acknowledge these concepts.

The proposal was inserted.

Discussion is too long. Several paragraphs can be deleted.

Discussion was shortened as per suggestion. Thank You for valuable corrections!

Thank You very much indeed.

Sincerely

Audrius Dulskas, MD, PhD

Reviewer 2 Report

The paper describes trends in liver cancer incidence and mortality in the period 1998-2015 in Lithuania using the national population-based cancer register data. The paper is clear written and has sufficient amount of details on sex-, age- and morphology-specific liver cancer rates with time. Did authors have an opportunity to assess if liver cancer incidence temporal patterns by liver cancer stage?  If so, how the trends look like?

Although the liver cancer incidence and mortality increased over the total follow-up period, there was a negative borderline significant (p=0.1) trend both for the incidence of about 6% per year in the period of 1998-2005 and for mortality of about 8% per year in the period of 1998-2001. What could be the reasons for that?    

Specific comments:

There is a one case discrepancy between the Abstract, line 118 and Table 1 in number of deaths due to liver cancer (2922 in the text vs 2923 in the table). Please clarify or correct.

Please provide axis titles for Figures 1 & 2.

Line 181: it should read as "HCV (18.7%)", please correct.

There is a number of abbreviations used without introducing them at first time such as CI, LAM, ETV. Please check that all the abbreviations are spelled out before using them. 

Line 233: Please put CHB in parentheses as an abbreviation for chronic hepatitis B.  

It is recommended to change the Y-scale to improve the visibility of liver cancer trends over time for Figures 3 & 4.   

Table 2. Why in the trend analysis of HCC incidence years 2006-2007 are missing, and in the mortality analysis year 2001 included both in Trend 1 (1998-2001) and Trend 2 (2001-2015) periods?     

Figure 4. What was the purpose of age standardisation when presenting age-specific incidence rates by calendar year? Same question for Table 2 for the incidence trends by age groups?

Author Response

First round

A point-by-point response to the reviewer's comments:

Dear Reviewers,

Thank you for your letter and constructive comments concerning our manuscript entitled “Trends in incidence and mortality of primary liver cancer in Lithuania 1998-2015”. The paper was revised substantially. Following changes have been made. They are as follows, revised paragraphs, sentences, words are below:

Reviewer #2: The paper describes trends in liver cancer incidence and mortality in the period 1998-2015 in Lithuania using the national population-based cancer register data. The paper is clear written and has sufficient amount of details on sex-, age- and morphology-specific liver cancer rates with time. Did authors have an opportunity to assess if liver cancer incidence temporal patterns by liver cancer stage?  If so, how the trends look like?

Unfortunately, in the Cancer Registry database only about 50% of liver cancer are reported with stage of disease, therefore, analysis of trends by stage makes no sense.

Although the liver cancer incidence and mortality increased over the total follow-up period, there was a negative borderline significant (p=0.1) trend both for the incidence of about 6% per year in the period of 1998-2005 and for mortality of about 8% per year in the period of 1998-2001. What could be the reasons for that?    

Trends in the periods 1998-2005 for HCC incidence and 1998-2001 for mortality should be interpreted as stable due to insignificant trend, also keeping in mind that changes in rates were exceedingly small – from 2.16 to 1.31 for incidence and from 4.02 to 3.30 for mortality.

Specific comments:

There is a one case discrepancy between the Abstract, line 118 and Table 1 in number of deaths due to liver cancer (2922 in the text vs 2923 in the table). Please clarify or correct.

The number corrected (true one is 2923).

Please provide axis titles for Figures 1 & 2.

Axis title provided.

Line 181: it should read as "HCV (18.7%)", please correct.

Corrected, thank you.

There is a number of abbreviations used without introducing them at first time such as CI, LAM, ETV. Please check that all the abbreviations are spelled out before using them. 

Abbreviations inserted.

Line 233: Please put CHB in parentheses as an abbreviation for chronic hepatitis B.  

Inserted – thank you very much.

It is recommended to change the Y-scale to improve the visibility of liver cancer trends over time for Figures 3 & 4.   

Figures changed accordingly.

Table 2. Why in the trend analysis of HCC incidence years 2006-2007 are missing, and in the mortality analysis year 2001 included both in Trend 1 (1998-2001) and Trend 2 (2001-2015) periods?     

Typo error is corrected 2008 replaced with 2005

Year 2001 included in Trend1 and Trend2 because significant change in trend occurred in 2001.

Figure 4. What was the purpose of age standardisation when presenting age-specific incidence rates by calendar year? Same question for Table 2 for the incidence trends by age groups?

Standardization is needed to avoid influence of changes in age structure in population for comparison of rates during time period. This approach is used also for rates in age groups.

Thank You very much indeed

Audrius Dulskas, MD, PhD